OBSERVATION

# Fecal Microbial Community Composition in Myeloproliferative Neoplasm Patients Is Associated with an Inflammatory State

Andrew Oliver,[a] Kenza El Alaoui,[b,c] Carolyn Haunschild,[d] Julio Avelar-Barragan,[a] Laura F. Mendez Luque,[e] Katrine Whiteson,[a] Angela G. Fleischman[b,e,f]

aDepartment of Molecular Biology and Biochemistry, School of Biological Sciences, University of California Irvine, Irvine, California, USA
bDivision of Hematology/Oncology, School of Medicine, University of California Irvine, Irvine, California, USA
cDepartment of Internal Medicine, Université Libre de Bruxelles, Brussels, Belgium
dDivision of Gynecologic Oncology, School of Medicine, University of California Irvine, Irvine, California, USA
eBiological Chemistry, School of Medicine, University of California Irvine, Irvine, California, USA
fChao Family Comprehensive Cancer Center, University of California Irvine, Irvine, California, USA

Andrew Oliver and Kenza El Alaoui contributed equally to this article. Author order was determined by seniority.

**ABSTRACT** The capacity of the human microbiome to modulate inflammation in the context of cancer is becoming increasingly clear. Myeloproliferative neoplasms (MPNs) are chronic hematologic malignancies in which inflammation plays a key role in disease initiation, progression, and symptomatology. To better understand the composition of the gut microbiome in patients with MPN, triplicate fecal samples were collected from 25 MPN patients and 25 non-MPN controls. Although most of the variance between the microbial community compositions could be attributed to the individual (permutational analysis of variance [PERMANOVA], $R^2 = 0.92$, $P = 0.001$), 1.7% of the variance could be attributed to disease status (MPN versus non-MPN). When a more detailed analysis was performed, significantly fewer reads mapping to a species of *Phascolarctobacterium*, a microbe previously associated with reduced inflammation, were found in MPNs. Further, our data revealed an association between *Parabacteroides* and tumor necrosis factor alpha (TNF-$\alpha$), an inflammatory cytokine elevated in MPNs. Taken together, our results indicate a significant difference in the microbiome of MPN patients compared to non-MPN controls, and we identify specific species which may have a role in the chronic inflammation central to this disease.

**IMPORTANCE** MPNs are chronic blood cancers in which inflammation plays a key role in disease initiation, progression, and symptomatology. The gut microbiome modulates normal blood development and inflammation and may also impact the development and manifestation of blood cancers. Therefore, the microbiome may be an important modulator of inflammation in MPN and could potentially be leveraged therapeutically in this disease. However, the relationship between the gut microbiome and MPNs has not been defined. Therefore, we performed an evaluation of the MPN microbiome, comparing the microbiomes of MPN patients with healthy donors and between MPN patients with various states of disease.

**KEYWORDS** cytokines, inflammation, microbiome, myeloproliferative neoplasm

There is an expanding appreciation for associations between the gut microbiome and hematopoiesis. Studies involving the microbiome in hematologic malignancies have primarily focused on acute myeloid leukemia and hematopoietic stem cell transplantation, specifically evaluating the impact of the microbiome on infection (1), hematopoietic reconstitution, and graft versus host disease (GVHD) (2). To date, no studies have investigated the gut microbiome of myeloproliferative neoplasm (MPN) patients.

Address correspondence to Angela G. Fleischman, agf@uci.edu.

The authors declare no conflict of interest.

MPN is a hematologic malignancy with a hallmark feature of chronic inflammation. The inflammation in MPN is multifactorial, and the neoplastic clone itself induces inflammation; however, chronic inflammation may precede the development of MPN and play a critical role in disease initiation. Disease manifestations are variable among MPN patients, even those with identical MPN driver mutations. This suggests that other forces modulating inflammation play an instructive role in MPN disease manifestation. We conducted a pilot study to test the hypothesis that the microbiome of MPN patients is distinct from controls and that changes in gut microbiome composition may be associated with MPN pathogenesis.

Twenty-five MPN patients and 25 controls participated in this pilot study, with inclusion and exclusion criteria shown in Table S1 in the supplemental material. At enrollment, all participants completed a survey which included questions about demographics, lifestyle, and other clinical covariates of interest (Table S2). The MPN cohort additionally completed questions on disease characteristics, treatment regimens, and symptom burden (Table S3).

Participants collected three fecal samples over the course of 1 week. We performed 16S rRNA gene sequencing and analysis with QIIME2, resulting in a table of 100% operational taxonomic units (OTUs) (see Supplemental Methods). To investigate whether having MPN was associated with changes in alpha diversity within the gut microbiome, we analyzed the number of distinct species (richness) and the distribution (evenness) of those species. Both richness and evenness did not significantly differ between MPN patients and controls (Fig. 1A), consistent with recent findings of Barone et al. (3) comparing the gut microbiome of polycythemia vera (PV) patients and healthy controls. The most abundant bacterial taxa found in this cohort came from the taxonomic families *Ruminococcaceae* (mean, 32.1%), *Lachnospiraceae* (mean, 26.7%), and *Bacteroidaceae* (mean, 21.7%) (Fig. 1B).

We next asked whether there were specific taxa that differed between patients with MPN and controls. A random forest model was capable of distinguishing between patients with MPN and controls by using microbiome composition alone (Fig. 1C). While several taxa informed the random forest model (Fig. 1D), we found that an OTU from the genus *Phascolarctobacterium* was critical in differentiating patients with MPN from controls. Furthermore, gut microbiomes from controls have significantly higher raw abundances of sequence reads mapping to *Phascolarctobacterium* (Fig. 1E). Using linear discriminant analysis to confirm the random forest results showing differential abundance of *Phascolarctobacterium* between patients with MPN and controls also revealed a significantly lower relative abundance of *Phascolarctobacterium* in patients with MPN (Fig. S1). We obtained a similar result from a traditionally built and cross-validated random forest model regarding the importance of *Phascolarctobacterium* in distinguishing MPN and healthy individuals. Increased *Phascolarctobacterium* is associated with benefits that include protection from *Clostridium difficile* infection (4) and lower levels of C-reactive protein (CRP) (5). Further, decreased abundance of *Phascolarctobacterium* is observed in autoimmune diseases such as primary sclerosing cholangitis and ulcerative colitis (6) and may be associated with decreases in the short-chain fatty acid propionate in the gut, which, in turn, can influence inflammation (7). *Phascolarctobacterium* may protect from inflammation; thus, lower *Phascolarctobacterium* in MPN patients corroborates a chronic inflammatory state in this disease.

Changes in taxonomic composition may indicate differences in the functional potential of microbial communities. We inferred gene composition from taxonomic composition (Supplemental Methods) and found that the microbiomes of MPN patients were enriched for genes involved in D-glucuronate metabolism (Fig. 1F). Changes in abundances of $\beta$-D-glucuronidases are associated with colon cancer and other inflammatory diseases (8).

The taxonomic composition of gut microbiomes within this cohort was largely personalized (PERMANOVA, $R^2 = 0.65$, $P = 0.001$), reflecting the individualistic nature of the microbiome. An MPN diagnosis explained 1.7% (PERMANOVA, $P = 0.001$) of the between-cohort variance in the microbiome (Fig. 1G), suggesting subtle but significant

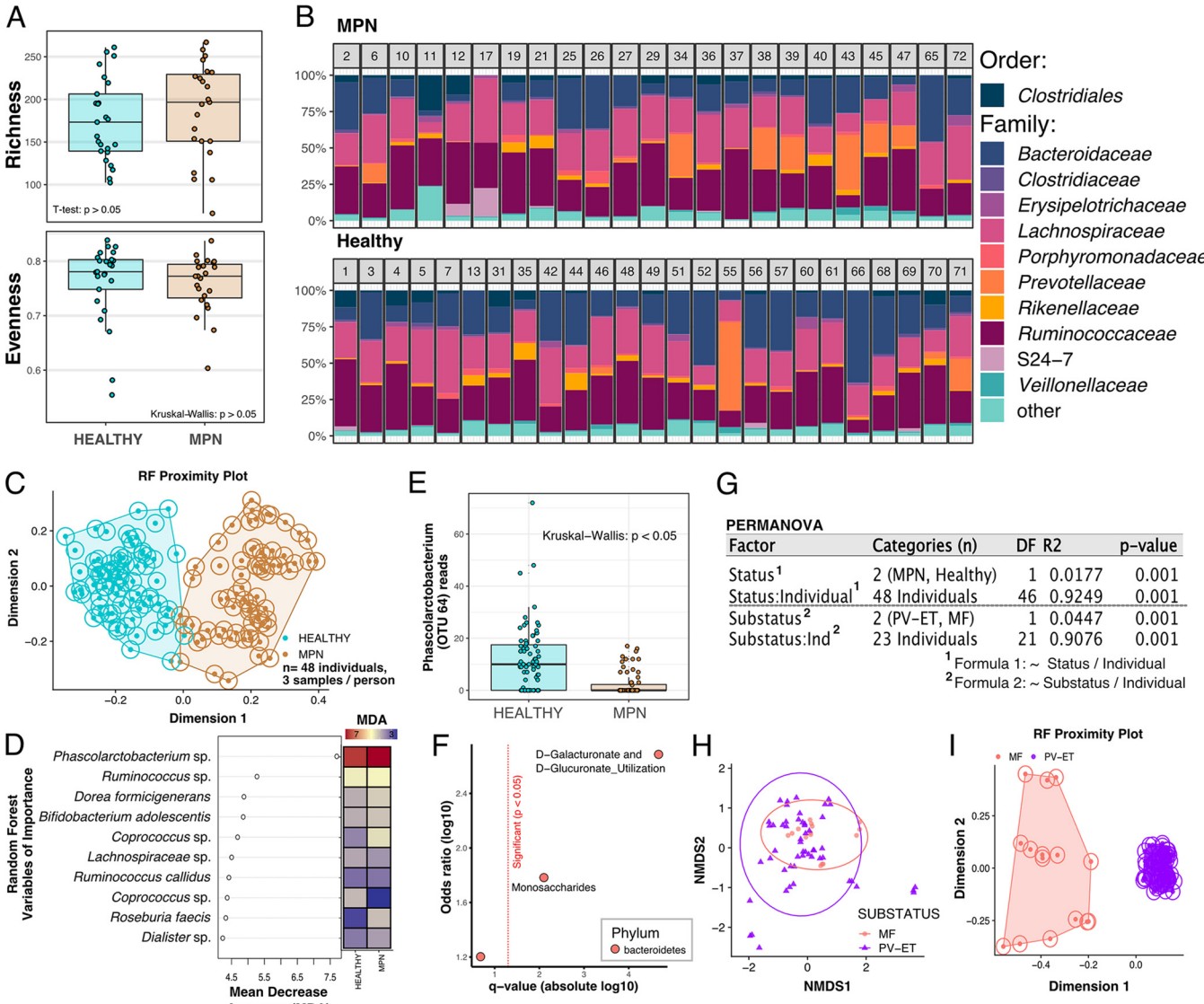

**FIG 1** Characterization of the gut microbiome in patients with myeloproliferative neoplasms. (A) Alpha diversity was averaged within the individual, showing no significant differences between health status for richness (number of distinct operational taxonomic units [OTU]) and evenness (distribution of those species). (B) Gut microbial families in MPN (top) and non-MPN (bottom) subjects averaged within the individual (numbers on top of bars). (C and D) Permutated random forest plot of all samples from MPN and non-MPN individuals (C), identifying taxa (D) that were indicative of health status. (E) Normalized number of reads mapping to *Phascolarctobacterium* spp. from all samples of MPN patients and non-MPN individuals. (F) Use of phylogenize to identify functional potential of the communities enriched among MPN patients. (G) PERMANOVA results showing the significance and variance in microbiome composition explained by each tested factor. (H) Unsupervised ordination of the microbiomes from patients with PV and ET versus MF. (I) Random forest proximity plot distinguishing MPN substatus based on the gut microbial community. NMDS, nonmetric multidimensional scaling.

differences between the microbiomes of patients with MPN from controls. To contextualize this variation, consider the extreme intervention of ileocecal resection in Crohn's disease, which explained 5% of microbiome variance (9). Other factors, such as diet and cohabitation, are known to shape microbiome composition. In the present cohort, approximately half of the study participants were cohabitants, including 9 MPN patient-normal pairs and 3 healthy-healthy pairs, comprising 12 different households. Of the MPN patients and non-MPN subjects cohabitating, living together explained 50% of the variance in the microbiome (Fig. 1G), consistent with reports (10) showing that cohabiting people usually have the same diet, hygiene, and lifestyle, all of which strongly affect microbiome composition.

Since MPNs can be stratified into subtypes based on phenotype, we sought to determine if there were specific microbial signatures between the subtypes measured

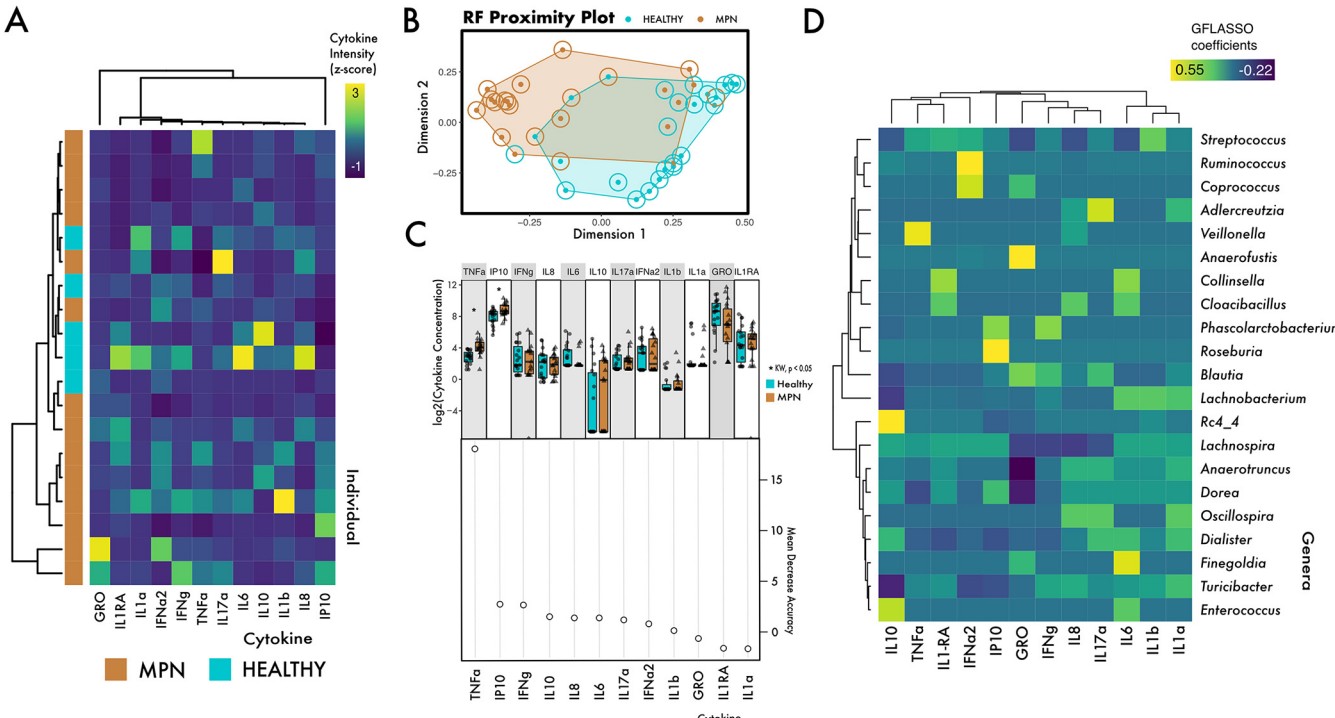

**FIG 2** Cytokines and the microbiome in MPNs. (A) Heatmap of plasma cytokine concentrations in a subset of MPN patients and additional controls with cytokines scaled using Z-scores. (B) Random forest plot utilizing cytokine profile to distinguish MPNs from non-MPNs. Dots represent the actual health status, and circles around the dots represent the RF classification. (C) Cytokines that were relied upon most heavily to make the classification of MPNs versus normal, particularly TNF-$\alpha$. (D) Grid-fused least absolute shrinkage and selection operator (LASSO) regression to select microbes that best predicted cytokine abundances in MPN patients identified several OTUs that may have correlative relationships with various cytokines.

in our cohort. Unsupervised ordination analysis of MPN subtypes showed that microbiomes of myelofibrosis (MF) patients had more similar community composition than those from PV or essential thrombocythemia (ET) (Fig. 1H). However, when the analysis was performed using a supervised random forest (RF) approach, distinct differences between early- (PV/ET) and late-stage (MF) MPNs were observed (Fig. 1I). Moreover, clustering from PV and ET patients was dense, whereas that from MF was more dispersed. Dysbiotic individuals with a larger spread in microbial community composition than non-MPN individuals have been called the "Anna Karenina principle" for animal microbiomes (11), paralleling Leo Tolstoy's dictum that "all happy families look alike; each unhappy family is unhappy in its own way." These data suggest that early- and late-stage subtypes of MPNs might be differentiated by the composition of gut microbes.

We sought to correlate plasma cytokines with microbial composition in the MPN cohort. We had available plasma from 20 individuals, which included 15 MPN patients and 5 controls (Fig. 2A); we measured 12 cytokines in all available samples. We found increased plasma concentrations of tumor necrosis factor alpha (TNF-$\alpha$) and interferon gamma-inducible 10-kDa protein (IP10) in MPN patients (Fig. 2C), consistent with other studies (12). Notably, although a random forest model could clearly distinguish MPN and non-MPN using the microbiome (Fig. 1E), the classification suffered considerably when using only cytokine concentration (Fig. 2B). The largest contributing factor to the cytokine RF model was TNF-$\alpha$ (Fig. 2C), which plays a critical role in MPN pathogenesis by creating an environment that is conducive to the growth of the neoplastic clone (12). Integrating the microbiome and cytokine data revealed associations between cytokine-taxa pairs (Fig. 2D). We found an association between TNF-$\alpha$ and the genus *Veillonella* (Fig. 2D). *Veillonella* stimulates TNF-$\alpha$ production of peripheral blood mononuclear cells in a dose-dependent fashion (13). Species of *Veillonella* have also been implicated in Crohn's disease (14), highlighting a potential role in inflammation. Using Spearman correlations as an alternative way to investigate TNF-$\alpha$ and the microbiome,

we found that the genus *Parabacteroides* may also be associated with TNF-$\alpha$ production in MPN; however, this trend was not significant after correcting for multiple comparisons ($r = 0.65$, $P = 0.016$, $q = 0.77$) (Fig. S2). Interestingly, one study found a species of *Parabacteroides* to be enriched in patients with colorectal carcinoma (15). Further, *Parabacteroides* abundance was negatively correlated with intake of fruits and vegetables. Conceivably, dietary nutrients such as vitamins and fiber may be an important covariate in the management of inflammatory diseases such as MPN. For example, low-fiber diets are associated with colonic inflammation and can be lessened by switching to a high-fiber diet, coincident with changes in colonic microbial metabolism (16). Future studies should examine how dietary interventions in MPN patients could be helpful to reduce inflammation, in part via modulation of the microbiome.

It is difficult to distinguish whether the microbiome affects MPN initiation and symptoms or whether the inflammatory response to the microbiome is exaggerated in MPN, as it is widely influenced by multiple factors. A potential caveat of this study is the heterogeneity of treatments for the individuals with MPN; indeed, we suspect that different medications will have variable effects on the microbiome. However, due to the pilot nature of this initial investigation, we are unable to confidently determine these differences, and we surmise that this is an important avenue for future research. Furthermore, therapeutic manipulation of the microbiome using diet, probiotics, or potentially fecal transfer with the intent of reducing inflammation in MPN remains unexplored. Despite the limitations of this initial pilot study, this study is an important step in the path to better understanding the role of the microbiome in MPN.

**Data availability.** The code used for the statistical analysis can be found on GitHub under the repository https://github.com/aoliver44/MPN_project. A Dockerfile is provided to reproduce the environment and packages necessary for the code. All sequence data can be found under BioProject accession no. PRJNA795185.

## SUPPLEMENTAL MATERIAL

Supplemental material is available online only.
**SUPPLEMENTAL FILE 1**, PDF file, 0.9 MB.

## ACKNOWLEDGMENTS

We acknowledge the T32 training grant from UC Irvine's training program in microbiology and infectious diseases, which supported A.O. (1T32AI14134601A1), and T32 CA060396 from UC Irvine's training program in gynecologic oncology, which supported C.H. We also acknowledge the UCI Microbiome Center for granting a pilot award, providing consulting on study design, and supporting the generation of the microbiome amplicon sequence data. We also acknowledge the UCI Genomic High Throughput Facility, a Chao Family Comprehensive Cancer Center Shared Resource supported by P30CA062203 (Cancer Center support grant). We would also like to express enormous gratitude to the study participants who donated time and energy, which made the study possible.

A.O., J.A.-B., and C.H. analyzed data and wrote the paper; K.E.A. collected data, analyzed results, and wrote the paper; L.F.M.L. collected data; K.W. analyzed data and wrote the paper; and A.G.F. conceived the study, analyzed results, and wrote the paper.

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
