## [Reviewer comments · Microbiology Spectrum]

Microbiology Spectrum

Fecal microbial community composition in myeloproliferative neoplasm patients is associated with an inflammatory state

Andrew Oliver, Kenza El Alaoui, Carolyn Haunschild, Julio Avelar-Barragan, Laura Mendez Luque, Katrine Whiteson, and Angela Fleischman

Corresponding Author(s): Angela Fleischman, University of California, Irvine

Review Timeline:

Submission Date:	January 7, 2022
Editorial Decision:	February 15, 2022
Revision Received:	March 30, 2022
Accepted:	April 11, 2022

Editor: Jan Claesen

Reviewer(s): Disclosure of reviewer identity is with reference to reviewer comments included in decision letter(s). The following individuals involved in review of your submission have agreed to reveal their identity: Yun Du (Reviewer #2); Abigail Esenam Asangba (Reviewer #4)

Transaction Report:

DOI: <https://doi.org/10.1128/spectrum.00032-22>

February 15, 2022

Dr. Angela G Fleischman
University of California, Irvine
Medicine
839 Medical Sciences Ct
Sprague Hall 126
Irvine, CA 92617

Re: Spectrum00032-22 (Fecal microbial community composition in myeloproliferative neoplasm patients is associated with an inflammatory state)

Dear Dr. Angela G Fleischman:

Thank you for submitting your research to Spectrum. The Reviewers agree that your manuscript is well-written and that the methods used seem well performed. They did bring up several comments and suggestions for improving the paper and I would be happy to consider a revised version in which these Reviewer points have been carefully addressed. Also, please make the data in BioProject PRJNA795185 available to the reviewers. This can be done either (i) by making it public (and I would understand if you have reservations doing so before the manuscript is published), or (ii) by providing a token that the reviewers can use to access the data.

Link Not Available

Sincerely,

Jan Claesen

Journals Department
Reviewer comments:

Reviewer #1 (Public repository details (Required)):

1. Code used for the statistical analysis can be found on GitHub under the repository: https://github.com/aoliver44/MPN_project.
2. PRJNA795185 is listed as the data BioProject but it doesn't appear when searching at NCBI.

Reviewer #1 (Comments for the Author):

Thank you for sharing this interesting observation. The manuscript is well-written, and the results are clear and well-explained.

This was a small-scale pilot study. The author found three bacteria associated with the pathogenesis of myeloproliferative neoplasms (MPNs). The first one was *Phascolarctobacterium*, which was decreased in MPN patients. The other two were from the genus *Veillonella* and *Parabacteroides*, which were potential pathobionts that positively correlated with the increase of proinflammatory TNF alpha. Although it is hard to distinguish a causal relationship between fecal bacteria and MPNs, this study still provides information about the association of the existence of proinflammatory bacteria in MPN patients.

Comments:

1. I think that some modification is needed in the Visual Abstract. The "50 X 3 fecal sample => 16S sequencing => result figures" was not done in parallel with "15 MPN + 5 Healthy Blood sample => cytokine => result figures". The latter one should be placed after the former (as the further analysis).
2. Please write out MPN, PV, and MF in full in the Visual Abstract.
3. In Figure 1 (A), why was ANOVA used for the analysis of two groups?
4. In Figure 1 (B), I recommend arranging the list of bacteria as Order: Clostridiales; Family: other bacteria; and remove the case number on the top of the bar (replace with this sentence: "each bar represents an individual subject").
5. In Figure 2 (C), why was cytokine "abundance" used instead of concentration?
6. In Figure 2 (D), please indicate the bacterial taxonomy on the right axis.
7. In the Supplemental Methods, Microbiome sequencing, Illumina Miseq usually targets the V3-V4 region of rRNA. Why was the V4-V5 region targeted in PCR amplification in the current study?
8. Line 74, change the "Twenty-five MPN patients and 25 controls" to "25 MPN patients and 25 controls".
9. Line 75, add a comma to the sentence "At enrollment, all participants".
10. Line 182, add a comma to the sentence "study, this study".
11. PRJNA795185 is listed as the data BioProject but it doesn't appear when searching at NCBI.

Reviewer #2 (Public repository details (Required)):

Please provide the link to the sequence data. I couldn't find the data following BioProject PRJNA795185.

Reviewer #2 (Comments for the Author):

Methods:

Study population:

How were the cases and control selected? For example, how many MPN cases are there in the MPH cohort? Then how many were included based on the inclusion criteria and how many were excluded based on your exclusion criteria? After the cases met the prerequisite of inclusion and exclusion criteria, did you select the cases randomly?

Please explain how the controls were selected.

Sampling

L81, it was mentioned that the fecal samples were collected over the course of one week. How did it collect? Did the participants have prebiotics before sampling? When and how were the fecal sample sampled?

Sequence

Did you have simultaneously negative control when PCR and 16s rRNA sequencing?

In supplement methods, it mentioned two positive control mock communities were used. What organisms were the two positive controls?

Pre-processing

In the supplement method, it mentioned the lowest number of reads per sample was 3782 and the average was 81.081. Before assignment with taxonomic classification, did you filter out some samples with insufficient reads or include all the sequenced samples? If the reads of one sample are too low, the sampling or amplification might have failed and the reads only come from some contamination.

Please clarify which version of QIIME2 was used.

Statistical methods, results, and discussion

For Figure 1A, please explain why ANOVA and Kruskal-Wallis were used for richness and evenness.

In line 94, 'we found an operational taxonomic unit (OTU) from the genus *Phascolarctobacterium* was critical in differentiating patients with MPN from controls.' There were thousands of OTUs, how did the OTU from the genus *Phascolarctobacterium* stand out? Did you apply the Kruskal-Wallis thousands of times? If so, we should use adjusted p values due to the high false-positive rate resulting from multiple tests.

PERMANOVA is a method to check whether there is a difference between communities (beta diversity.) It is recommended to check the global community before looking into some specific organism. Thus, it is suggested to put the PERMANOVA results

before the random forest.

In line 146, you subset 20 individuals for cytokines analysis, please describe how the 20 individuals were subset.

In Figure 2D, LASSO regression was used to select microbes that best predicted cytokine abundance. Were the organisms shown in figure 2D overlapped or combined for each cytokine?

Data availability

Please provide the link to the sequence data. I couldn't find the data following BioProject PRJNA795185.

Additional comment

Each subject had three fecal samples? Was there any difference in microbiome among the three samples from one subject?

Supplement Table 3:

There are some abbreviations in table 3. Please add the original name under Supplement table3.

Reviewer #3 (Comments for the Author):

This article investigated the relationship between the gut microbiome and myeloproliferative neoplasms (MPN) using 50 fecal samples where 25 samples are from patients with MPN and 25 from non-MPN controls. Major findings are a significant difference in the gut microbiome structure between MPN and non-MPN and a set of differentially abundant bacterial species between MPN and non-MPN that may have a role in the chronic inflammation.

Major comments:

-. Authors claims that this study is the first evaluation of the MPN microbiome. However, a quick google search identified the following papers. Authors should discuss about the existing findings in comparison with ones in the manuscript.

Investigating the role of the gut microbiome in the inflammatory state of myeloproliferative neoplasms

<https://ashpublications.org/blood/article/132/Supplement%201/3051/263913/Investigating-the-Role-of-the-Gut-Microbiome-in>

An abnormal host/microbiomes signature of plasma-derived extracellular vesicles is associated to polycythemia vera

<https://www.readcube.com/articles/10.3389/fonc.2021.715217>

Impact of host, lifestyle and environmental factors in the pathogenesis of MPN

<https://www.ncbi.nlm.nih.gov/pmc/articles/PMC7463688/>

-. Authors used richness and evenness to measure alpha diversity. Both of them do not consider phylogenetic relationship in measuring alpha diversity. Authors should include measures (for example, phylogenetic diversity index) which consider phylogenetic relationships of OTUs.

-. As PERMANOVA results indicated, inter-individual difference within the groups is huge. Authors pointed out a couple of confounding factors (for example, "Living together") and have metadata included in Supplementary. I don't see information of how these confounding factors were adjusted in PERMANOVA analysis and when markers were identified.

-. In Supplementary, please include a plot from unsupervised ordination analysis annotated with information of MPN (PV, ET, MF) and non-MPN and biological replicates to see variation between biological replicates and between MPN and non-MPN.

Minor comments:

-. Random Forest approach was used to identify differentially abundant bacteria and cytokines. Please describe how random forest models built were correctly validated.

-. Figure 1C, Figure 1I, and Figure 2B: please include information of how many markers detected by Random Forest were used.

-. It seems that authors used different approaches for ordination analysis. Please describe which approaches were used.

- Line 118: I don't see PERMANOVA, R2=0.65 in Figure 1G.
- Line 117-120: Are PERMANOVA R2 values from different models comparable?
- Figure 1D: please include description of color.
- Figure 1H: please use separate color and shape for PV and ET
- Figure 1I: Were markers identified by Random Forest with 3 groups (PV, ET, MF) used?
- Line 159: what is an undirected spearman?

Reviewer #4 (Comments for the Author):

Summary:

The study reported in "Fecal microbial community composition in myeloproliferative neoplasm patients is associated with an inflammatory state" by Oliver and colleagues investigated the relationship between the gut microbiome and myeloproliferative neoplasms (MPN). Their results show that while most of the variance can be explained by individual differences, 1.7% of the variance could be attributed to disease status (MPN vs non-MPN). They also identified specific species which may have a role in the chronic inflammation central to this disease.

While I believe the main objective of this work is important and provides a much needed albeit preliminary study on the role of the gut microbiome in MPN, and is potentially publishable in the Spectrum journal, I have some concerns I would like the authors to address prior to publication.

Importance:

L53: Authors should use "Healthy" instead of "normal".

OTU vs ASV:

Clustering amplicons into operational taxonomic units (OTUs) has given way to amplicon sequence variants (ASVs) which can provide more significant advantage and more precise identification of microbes. The authors should consider classifying amplicons into ASVs instead of OTUs using the most current version of QIIME 2 or DADA 2.

Database:

Greengenes has also not be updated in a while, I believe since 2013. The authors should consider using databases which are continuously updated such as SILVA, RDP or GTDP.

L126: Authors should use "Healthy" instead of "normal".

Staff Comments:

Preparing Revision Guidelines

- Point-by-point responses to the issues raised by the reviewers in a file named "Response to Reviewers," NOT IN YOUR COVER LETTER.

- Upload a compare copy of the manuscript (without figures) as a "Marked-Up Manuscript" file.
- Each figure must be uploaded as a separate file, and any multipanel figures must be assembled into one file.
- Manuscript: A .DOC version of the revised manuscript
- Figures: Editable, high-resolution, individual figure files are required at revision, TIFF or EPS files are preferred

Please return the manuscript within 60 days; if you cannot complete the modification within this time period, please contact me. If you do not wish to modify the manuscript and prefer to submit it to another journal, please notify me of your decision immediately so that the manuscript may be formally withdrawn from consideration by Microbiology Spectrum.

Thank you for sharing this interesting observation. The manuscript is well-written, and the results are clear and well-explained.

This was a small-scale pilot study. The author found three bacteria associated with the pathogenesis of myeloproliferative neoplasms (MPNs). The first one was *Phascolarctobacterium*, which was decreased in MPN patients. The other two were from the genus *Veillonella* and *Parabacteroides*, which were potential pathobionts that positively correlated with the increase of proinflammatory TNF alpha. Although it is hard to distinguish a causal relationship between fecal bacteria and MPNs, this study still provides information about the association of the existence of proinflammatory bacteria in MPN patients.

Comments:

1. I think that some modification is needed in the Visual Abstract. The “50 X 3 fecal sample => 16S sequencing => result figures” was not done in parallel with “15 MPN + 5 Healthy Blood sample => cytokine => result figures”. The latter one should be placed after the former (as the further analysis). It should be more like this:

2. Please write out MPN, PV, and MF in full in the Visual Abstract.
3. In Figure 1 (A), why was ANOVA used for the analysis of two groups?
4. In Figure 1 (B), I recommend arranging the list of bacteria as Order: Clostridiales; Family: other bacteria; and remove the case number on the top of the bar (replace with this sentence: “each bar represents an individual subject”).
5. In Figure 2 (C), why was cytokine “abundance” used instead of concentration?
6. In Figure 2 (D), please indicate the bacterial taxonomy on the right axis.
7. In the Supplemental Methods, Microbiome sequencing, Illumina Miseq usually targets the V3-V4 region of rRNA. Why was the V4-V5 region targeted in PCR amplification in the current study?
8. Line 74, change the “Twenty-five MPN patients and 25 controls” to “25 MPN patients and 25 controls”.
9. Line 75, add a comma to the sentence “At enrollment, all participants”.
10. Line 182, add a comma to the sentence “study, this study”.

11. PRJNA795185 is listed as the data BioProject but it doesn't appear when searching at NCBI.

Point by point responses to Reviewer's comments for "Fecal microbial community composition in myeloproliferative neoplasm patients is associated with an inflammatory state," (Spectrum00032-22) by Oliver et al.

Reviewer #1 (Public repository details (Required)):

1. Code used for the statistical analysis can be found on GitHub under the repository:
https://github.com/aoliver44/MPN_project.

2. PRJNA795185 is listed as the data BioProject but it doesn't appear when searching at NCBI.

Response: We will release upon publication (release date set for May 1, 2022 currently), however, for the review process we have made an NCBI SRA reviewer link:
<https://dataview.ncbi.nlm.nih.gov/object/PRJNA795185?reviewer=vitph9lka0nlii81e1givr1hps>

Reviewer #1 (Comments for the Author):

Thank you for sharing this interesting observation. The manuscript is well-written, and the results are clear and well-explained.

Response: Thank you, it was a team effort and we are eager to share our findings with the scientific community.

This was a small-scale pilot study. The author found three bacteria associated with the pathogenesis of myeloproliferative neoplasms (MPNs). The first one was Phascolarctobacterium, which was decreased in MPN patients. The other two were from the genus Veillonella and Parabacteroides, which were potential pathobionts that positively correlated with the increase of proinflammatory TNF alpha. Although it is hard to distinguish a causal relationship between fecal bacteria and MPNs, this study still provides information about the association of the existence of proinflammatory bacteria in MPN patients.

Response: Thank you, we feel that studies with human samples play an important role in understanding the role of the microbiome in health, although it is harder to determine causal relationships in observational studies.

1. I think that some modification is needed in the Visual Abstract. The "50 X 3 fecal sample => 16S sequencing => result figures" was not done in parallel with "15 MPN + 5 Healthy Blood sample => cytokine => result figures". The latter one should be placed after the former (as the further analysis).

Response: We have edited the visual abstract to make our study design more clear.

2. Please write out MPN, PV, and MF in full in the Visual Abstract.

Comments: Thank you, we have modified the visual abstract to reflect this.

3. In Figure 1 (A), why was ANOVA used for the analysis of two groups?

Response: Thank you. The underlying math between an ANOVA and t.test should be identical. In our results, both tests resulted in a $p = 0.43$. However, we have analyzed the two groups using a T-test and made the change ("ANOVA" to "T-test") in Figure 1A for clarity.

4. In Figure 1 (B), I recommend arranging the list of bacteria as Order: Clostridiales; Family: other bacteria; and remove the case number on the top of the bar (replace with this sentence: "each bar represents an individual subject").

Response: With regards to the case number - we believe this is valuable information to retain. This information will allow readers to go to our raw data for specific participants of interest.

5. In Figure 2 (C), why was cytokine "abundance" used instead of concentration?

Response: Thank you for pointing this out. Figure 2C Y axis has been changed to cytokine concentration, and text (line 151) was also corrected from abundance to concentration.

6. In Figure 2 (D), please indicate the bacterial taxonomy on the right axis.

Response: Bacteria genera was added as a label header on the right axis.

7. In the Supplemental Methods, Microbiome sequencing, Illumina Miseq usually targets the V3-V4 region of rRNA. Why was the V4-V5 region targeted in PCR amplification in the current study?

Response: We targeted the V4-V5 region because we were following the Earth Microbiome Project protocols, which are widely used for 16S rRNA gene analyses with large databases available for analysis and comparison. Using these primers also potentially allow for future comparisons across other studies (meta-analyses).

8. Line 74, change the "Twenty-five MPN patients and 25 controls" to "25 MPN patients and 25 controls".

Response: The 25 begins a sentence, it is our understanding that numbers should be spelled out if at the beginning of a sentence.

9. Line 75, add a comma to the sentence "At enrollment, all participants".

Done

10.Line 182, add a comma to the sentence "study, this study".

Done

11.PRJNA795185 is listed as the data BioProject but it doesn't appear when searching at NCBI.

The NCBI link is:

<https://dataview.ncbi.nlm.nih.gov/object/PRJNA795185?reviewer=vitph9lka0nlii81e1givrlhps>

Reviewer #2 (Public repository details (Required)):

Please provide the link to the sequence data. I couldn't find the data following BioProject PRJNA795185.

The NCBI reviewer link is:

<https://dataview.ncbi.nlm.nih.gov/object/PRJNA795185?reviewer=vitph9lka0nlii81e1givrlhps>

Reviewer #2 (Comments for the Author):

Methods:

Study population:

How were the cases and control selected? For example, how many MPN cases are there in the MPN cohort? Then how many were included based on the inclusion criteria and how many were excluded based on your exclusion criteria? After the cases met the prerequisite of inclusion and exclusion criteria, did you select the cases randomly?

Response: Cases were recruited from the hematology/oncology clinic of UC Irvine Health. UC Irvine Health providers screened their patient panels for candidates that fit entry criteria. Then, these patients were told about the study, asked additional screening questions to confirm eligibility, and asked to participate. All MPN patients who signed informed consent and gave stool samples were used as cases.

Please explain how the controls were selected.

Response: In some cases healthy controls accompanying the patient at the visit, such as spouses, were asked whether they would be interested in serving as healthy controls. Other healthy controls were recruited through word of mouth from the UC Irvine community.

Sampling

L81, it was mentioned that the fecal samples were collected over the course of one week. How did it collect? Did the participants have prebiotics before sampling? When and how were the fecal sample sampled?

Response: We added further explanation in Supplemental Methods (included below):

Stool Samples:

Participants were given three DNA/RNA shield fecal collection tubes (cat #R1101-E, Zymo Research, Irvine, CA) and three plastic stool collection receptacles to place on the toilet. The fecal collection tubes include a scoop on the lid, participants were instructed to scoop up a pea sized piece of stool, place it in the 9ml of DNA/RNA shield, and mix by inverting 10 times, and keep at room temperature. Participants were instructed that each sample should be collected from a distinct bowel movement (ideally on separate days). Participants were not given specific instructions on time of day the sample should be collected, nor given specific instructions on what to eat prior to sample collection. After all samples were collected from the entire cohort DNA was isolated with a ZymoBIOMICS DNA Miniprep Kit according to manufacturer's protocol.

Sequence

Did you have simultaneously negative control when PCR and 16s rRNA sequencing?

Response: Negative PCR control, did not amplify and was not included in the sequencing pool. Two positive controls (community standards) were used. They matched each other well and contained the taxa outlined in the standard.

In supplement methods, it mentioned two positive control mock communities were used. What organisms were the two positive controls?

Response: The mock communities used as positive controls were purchased from Zymo Research (Catalog #D6305). This is a DNA standard from a synthetic microbial community with a known composition. More information about the mock community composition can be found here: https://files.zymoresearch.com/protocols/d6305_d6306_zymbiomics_microbial_community_dna_standard.pdf

Pre-processing

In the supplement method, it mentioned the lowest number of reads per sample was 3782 and the average was 81.081. Before assignment with taxonomic classification, did you filter out some samples with insufficient reads or include all the sequenced samples? If the reads of one sample are too low, the sampling or amplification might have failed and the reads only come from some contamination.

Response: We did not filter prior to taxonomic assignment. Rarefaction would drop samples below the rarefaction limit (i.e., samples that did not amplify properly). We do not suspect contamination because the positive controls were clean and contamination tends to get magnified, so the taxonomic

composition would stand out among individuals' biological replicates analyzed. Individuals microbiomes were highly consistent. These observations lead us to believe that we likely don't have appreciable contamination in our data.

Please clarify which version of QIIME2 was used.

Response: Qiime2 2018.4

Statistical methods, results, and discussion

For Figure 1A, please explain why ANOVA and Kruskal-Wallis were used for richness and evenness.

Response: We used ANOVA because we were already looking at the effect of MPN subtype (3 groups) on alpha diversity, but we presented on the overall MPN vs Healthy (2 groups). Since the underlying math between an ANOVA and a T-test is essentially the same for 2 groups (indeed $F = t^2$), we left the ANOVA as is. However, we changed it to a t-test in text and in the figure, though expectedly, there was no change in p-value with the change in test.

A Kruskal-Wallis, a non-parametric test, was used because the assumption of normality was not met for evenness.

In line 94, 'we found an operational taxonomic unit (OTU) from the genus Phascolarctobacterium was critical in differentiating patients with MPN from controls.' There were thousands of OTU, how did the OTU from the genus Phascolarctobacterium stand out? Did you apply the Kruskal-Wallis thousands of times? If so, we should use adjusted p values due to the high false-positive rate resulting from multiple tests.

Response: The OTU was the top OTU identified by the permuted random forest (Figure 1D). We merely presented the raw data and a traditional statistical test to show "why" the RF identified OTU64 as good at distinguishing MPN and Healthy.

PERMANOVA is a method to check whether there is a difference between communities (beta diversity.) It is recommended to check the global community before looking into some specific organism. Thus, it is suggested to put the PERMANOVA results before the random forest.

Response: Thank you for the suggestion. Since we presented a taxonomy (in Figure 1B), we thought a natural next question was "What is taxonomically different between these 2 groups". While we agree with the natural flow of analyses that you listed, in an effort to be extremely concise for this 1200 word observation, we answered these questions slightly out of order.

In line 146, you subset 20 individuals for cytokines analysis, please describe how the 20 individuals were subset.

Response: We had plasma available from 20 people, we used all plasma samples available to us. This was clarified in the text in line 146.

In Figure 2D, LASSO regression was used to select microbes that best predicted cytokine abundance. Were the organisms shown in figure 2D overlapped or combined for each cytokine?

Response: They were combined. A similar analysis can be found by Liam et al (2018) DOI: 10.1111/tpj.13833, who also wrote the R package used. From their publication: "GFLASSO allowed us to select genes that jointly predict multiple related traits....in a single framework".

Data availability

Please provide the link to the sequence data. I couldn't find the data following BioProject PRJNA795185.

<https://dataview.ncbi.nlm.nih.gov/object/PRJNA795185?reviewer=vitph9lka0nlii81e1givrlhps>

Additional comment

Each subject had three fecal samples? Was there any difference in microbiome among the three samples from one subject?

Response: Yes, but these differences were minor. A PERMANOVA analyzing the bacterial community composition under the main effect of an individual shows that 92% of the variance could be attributed to samples originating from the same individual (Figure 1G). This result suggests there are little compositional differences within individuals.

Supplement Table 3:

There are some abbreviations in table 3. Please add the original name under Supplement table 3.

Response: The abbreviations were replaced by full names.

Reviewer #3 (Comments for the Author):

This article investigated the relationship between the gut microbiome and myeloproliferative neoplasms (MPN) using 50 fecal samples where 25 samples are from patients with MPN and 25 from non-MPN controls. Major findings are a significant difference in the gut microbiome structure between MPN and non-MPN and a set of differentially abundant bacterial species between MPN and non-MPN that may have a role in the chronic inflammation.

Major comments:

-. Authors claims that this study is the first evaluation of the MPN microbiome. However, a quick google

search identified the following papers. Authors should discuss about the existing findings in comparison with ones in the manuscript.

Response: Thank you. Two of these references are from our group (conference poster and review article). The article " An abnormal host/microbiomes signature of plasma-derived extracellular vesicles is associated to polycythemia vera" focuses on the Polycythemia Vera subtype of MPN and the analysis was limited to alpha diversity. We lightened the language in the importance section to:

However, the relationship between the gut microbiome and MPN has not been defined. Therefore, we performed an evaluation of the MPN microbiome, comparing the microbiomes of MPN patients with healthy donors and between MPN patients with varying states of disease.

We also added a mention in line 87 that our findings that the richness and evenness of the MPN cohort as not different than the healthy donor cohort were consistent with the manuscript " An abnormal host/microbiomes signature of plasma-derived extracellular vesicles is associated to polycythemia vera".

Investigating the role of the gut microbiome in the inflammatory state of myeloproliferative neoplasms
<https://ashpublications.org/blood/article/132/Supplement%201/3051/263913/Investigating-the-Role-of-the-Gut-Microbiome-in>

This is our poster from the American Society of Hematology meeting describing our initial findings.

An abnormal host/microbiomes signature of plasma-derived extracellular vesicles is associated to polycythemia vera
<https://www.readcube.com/articles/10.3389/fonc.2021.715217>

As mentioned above this article focuses on Polycythemia Vera. We wrote our manuscript prior to this article being published, but we have updated our manuscript to now reference this recent article.

Impact of host, lifestyle and environmental factors in the pathogenesis of MPN
<https://www.ncbi.nlm.nih.gov/pmc/articles/PMC7463688/>

This is a review article authored by us that alludes to this current study.

- . Authors used richness and evenness to measure alpha diversity. Both of them do not consider phylogenetic relationship in measuring alpha diversity. Authors should include measures (for example, phylogenetic diversity index) which consider phylogenetic relationships of OTUs.

Response: We used richness and evenness as we suspected these are some of the more understandable metrics of alpha diversity, and an audience of MPN clinicians may get "number of distinct taxa" over phylogenetically corrected diversity which Unifrac provides. We checked our results against Faith's PD (mathematically related to Unifrac), and found no difference in alpha diversity. We have indicated this in text and provided a qiime2 output here.

- As PERMANOVA results indicated, inter-individual difference within the groups is huge. Authors pointed out a couple of confounding factors (for example, 'Living together') and have metadata included in Supplementary. I don't see information of how these confounding factors were adjusted in PERMANOVA analysis and when markers were identified.

Response: This was a pilot project, we set out to determine whether there were microbial differences between MPN patients. We considered the possibility of including additional factors in our analysis, but found that it was beyond the scope of this pilot study and the 1200 word limit. We discuss the cohabitation subset of individuals in our response to questions about line 117 below.

- In Supplementary, please include a plot from unsupervised ordination analysis annotated with information of MPN (PV, ET, MF) and non-MPN and biological replicates to see variation between biological replicates and between MPN and non-MPN.

Response: Thank you, we have now included that figure as Supp. Fig 2, here is the legend: "Supplemental Figure 2: NMDS ordination of Bray Curtis distances for all microbiome samples. Numbers indicate individual, color and shape represent substatus."

Minor comments:

- Random Forest approach was used to identify differentially abundant bacteria and cytokines. Please describe how random forest models built were correctly validated.

Response: Admittedly we were not using a random forest in a traditional sense to build a predictive model, rather we were interested in features that were predictive of MPN vs Healthy. We utilized the RF Permute package in R using default parameters to accomplish this, which permutes the response variable to calculate p-values for importance of VIPs.

In response to this suggestion, we did build a RF model using the sklearn library in python. For this model, we split the data into a train-test split, using 70% of our data (species-level OTU) for training purposes. Admittedly we have a small n, big p, however, many studies (especially pilot studies such as this one) do not have access to sample sizes that would satisfy ML engineers. Nevertheless, hyperparameters were tuned using a 5-fold stratified crossfold validation strategy, using a Bayesian search method to optimize the search of hyperparameter space. The best model was then fit to the entire training data, and tested on the 30% test data. The model was evaluated using Balanced Accuracy, ROC curves, and other metrics. The AUC ROC of our model was 0.91. To analyze feature importance, we used SHAP values, which are based on game theory and are used to analyze the outcome of games (model outcomes for a single observation). The "player" (of the game, a feature) that contributed most in distinguishing MPN vs Healthy was still OTU 64 (*Phascolarctobacterium*), the same model we obtained from the RFPermute and presented in figure 1.

We now note on line 105 in text that: "We obtained a similar result from a traditionally built and cross-validated random forest model regarding the importance of *Phascolarctobacterium* in distinguishing MPN and healthy individuals."

- Figure 1C, Figure 1I, and Figure 2B: please include information of how many markers detected by Random Forest were used.

Response: All species went into the rfPermute. Rather the outcome of all trees were used for the proximity scores. Samples that were closer together at the terminal end of trees had greater pairwise proximity scores and thus were plotted closer together in a multidimensional scaling plot (MDS).

- . It seems that authors used different approaches for ordination analysis. Please describe which approaches were used.

Response: Both NMDS (non-metric multidimensional scaling, Figure 1H) and random forest proximity plots (Figure 1C, I, Figure 2B) were used, mentioned in the figure legend/Axis label. We have updated the figure legends for increased clarity. Proximity plots are particularly useful because traditional unsupervised ordination (such as NMDS) based on distances such as Bray Curtis or Euclidean, are prone to noise which might mask potential subgroups. We believe using both; supervised approaches and unsupervised approaches both contribute different and useful information.

- . Line 118: I don't see PERMANOVA, R2=0.65 in Figure 1G.

Response: Thank you, this was a typo based on an earlier calculation based a formula with several additional factors. However, we lost many samples to include these factors, and we did not feel this was appropriate with our small pilot cohort. We have made sure the statistics match the figure.

- . Line 117-120: Are PERMANOVA R2 values from different models comparable?

Response: The PERMANOVA R2 values from different models are not comparable, especially because they include different numbers of samples and individuals. The trend that individual captures the most variance is consistent across models, however. Because they had different N's we kept the models separate (if we combined into a large model, the model would only describe the number of samples found in the smallest factor [cohabitation]). Others have presented single factor PERMANOVAs, especially useful for effect size (R2) values. We are largely just trying to illustrate a small but significant difference in microbial composition between MPN and healthy.

- . Figure 1D: please include description of color.

Response: Color means decrease in accuracy for each group. We added a label to the color bar.

- . Figure 1H: please use separate color and shape for PV and ET

Response: We combined PV and ET because these two diseases represent early stage MPN and MF represents late stage MPN.

- . Figure 1I: Were markers identified by Random Forest with 3 groups (PV, ET, MF) used?

Response: We did look into this but since this analysis effectively reduces our N in half, we didn't report our findings.

- . Line 159: what is an undirected spearman?

Response: we removed the word undirected

Reviewer #4 (Comments for the Author):

Summary:

The study reported in "Fecal microbial community composition in myeloproliferative neoplasm patients is associated with an inflammatory state" by Oliver and colleagues investigated the relationship between the gut microbiome and myeloproliferative neoplasms (MPN). Their results show that while most of the variance can be explained by individual differences, 1.7% of the variance could be attributed to disease status (MPN vs non-MPN). They also identified specific species which may have a role in the chronic inflammation central to this disease.

While I believe the main objective of this work is important and provides a much needed albeit preliminary study on the role of the gut microbiome in MPN, and is potentially publishable in the Spectrum journal, I have some concerns I would like the authors to address prior to publication.

Importance:

L53: Authors should use "Healthy" instead of "normal".

Done

OTU vs ASV:

Clustering amplicons into operational taxonomic units (OTUs) has given way to amplicon sequence variants (ASVs) which can provide more significant advantage and more precise identification of microbes. The authors should consider classifying amplicons into ASVs instead of OTUs using the most current version of QIIME 2 or DADA 2.

Response: To clarify, we are using OTUs at 100%, which means they are essentially ASVs. An OTU could be defined at many cut-offs, which we should have made clear in the text. We have now clarified that we defined the cutoff at 100%. Our analysis followed standard procedures with QIIME2, including the use of 100% OTUs. We would not expect a re-analysis with a newer version of QIIME2 or DADA2 to alter the findings, and we have made our workflow reproducible so that others could achieve the same findings.

Line 89-90: We performed 16S rRNA gene sequencing and analysis with QIIME2 resulting in a table of 100% Operational Taxonomic Units (OTUs) (see Supplemental Methods).

Database:

Greengenes has also not been updated in a while, I believe since 2013. The authors should consider using databases which are continuously updated such as SILVA, RDP or GTDP.

Response: We agree that, moving forward, taxonomy should be assessed with newer databases. However, Greengenes is widely used in contemporary publications. Version 13.8 of the Greengenes database is the latest and it is still the database recommended by QIIME2. Consider Figure 5, taken from Balvočiūtė and Huson, 2017 (<https://doi.org/10.1186/s12864-017-3501-4>). This figure shows how well the taxonomies of the SILVA, RDP, Greengenes (GG), OTT, and NCBI databases map to each other. The Greengenes database performs well in both strict and loose mapping cases. Ultimately, the authors

conclude that, “While we find that SILVA, RDP and Greengenes map well into NCBI, and all four map well into the OTT, mapping the two larger taxonomies [OTT and NCBI] on to the smaller ones is problematic.”

L126: Authors should use "Healthy" instead of "normal".

Done

April 11, 2022

Dr. Angela G Fleischman
University of California, Irvine
Medicine
839 Medical Sciences Ct
Sprague Hall 126
Irvine, CA 92617

Re: Spectrum00032-22R1 (Fecal microbial community composition in myeloproliferative neoplasm patients is associated with an inflammatory state)

Dear Dr. Angela G Fleischman:

Thanks for addressing the reviewers' comments and for making the data available. I would like to congratulate you on the acceptance of your manuscript in Microbiology Spectrum!

Your manuscript has been accepted, and I am forwarding it to the ASM Journals Department for publication. You will be notified when your proofs are ready to be viewed.

Sincerely,

Jan Claesen
Editor, Microbiology Spectrum
